# Integrated algorithm combining plasma biomarkers and cognitive assessments accurately predicts brain β-amyloid pathology

Fengfeng Pan[1,4], Yanlu Huang[1,4], Xiao Cai [2,4], Ying Wang[1], Yihui Guan[3], Jiale Deng[2], Dake Yang[2], Jinhang Zhu[2], Yike Zhao[2], Fang Xie[3,5 ✉], Zhuo Fang[2,5 ✉] & Qihao Guo[1,5 ✉]

## Abstract

**Background** Accurate prediction of cerebral amyloidosis with easily available indicators is urgently needed for diagnosis and treatment of Alzheimer's disease (AD).

**Methods** We examined plasma Aβ42, Aβ40, T-tau, P-tau181, and NfL, with *APOE* genotypes, cognitive test scores and key demographics in a large Chinese cohort ($N = 609$, aged 40 to 84 years) covering full AD spectrum. Data-driven integrated computational models were developed to predict brain β-amyloid (Aβ) pathology.

**Results** Our computational models accurately predict brain Aβ positivity (area under the ROC curves (AUC) = 0.94). The results are validated in Alzheimer's Disease Neuroimaging Initiative (ADNI) cohort. Particularly, the models have the highest prediction power (AUC = 0.97) in mild cognitive impairment (MCI) participants. Three levels of models are designed with different accuracies and complexities. The model which only consists of plasma biomarkers can predict Aβ positivity in amnestic MCI (aMCI) patients with AUC = 0.89. Generally the models perform better in participants without comorbidities or family histories.

**Conclusions** The innovative integrated models provide opportunity to assess Aβ pathology in a non-invasive and cost-effective way, which might facilitate AD-drug development, early screening, clinical diagnosis and prognosis evaluation.

## Plain language summary

The numbers of people with Alzheimer's disease are increasing. People with Alzheimer's disease have changes in the brain as well as cognitive impairment, which is when a person has difficulty remembering, learning, concentrating, or making decisions. Innovative medicines and new treatments all target people with early Alzheimer's disease. However, the methods used currently to diagnose Alzheimer's disease are expensive and can be unpleasant for patients. We studied Chinese people with no cognitive impairment, some cognitive decline, mild cognitive impairment, Alzheimer's disease and non-Alzheimer's disease dementia. We established a computational model that can predict the changes seen in the brain in people with Alzheimer's disease from information including results of blood and memory tests. This non-invasive and cost-effective approach might improve early identification of those with Alzheimer's disease.

[1] Department of Gerontology, Shanghai Jiao Tong University Affiliated Sixth People's Hospital, Shanghai, China. [2] Department of Data & Analytics, WuXi Diagnostics Innovation Research Institute, Shanghai, China. [3] PET Center, Huashan Hospital, Fudan University, Shanghai, China. [4] These authors contributed equally: Fengfeng Pan, Yanlu Huang, Xiao Cai. [5] These authors jointly supervised this work: Fang Xie, Zhuo Fang, Qihao Guo. ✉email: fangxie@fudan.edu.cn; fang_zhuo@wuxidiagnostics.com; qhguo@sjtu.edu.cn

Alzheimer's disease becomes an ever-growing burden on public health[1]. Most existing treatments and ongoing clinical trials of innovative medicines all appeared more effective on MCI or early Alzheimer's disease patients, especially those who had abnormal levels of brain Aβ[2,3]. Currently, cerebral accumulation of extracellular amyloid plaques can be detected by positron emission tomography (PET) scan or reflected by cerebrospinal fluid (CSF) Aβ measurements after lumbar puncture[4]. However, the high cost and accessibility limitations of PET images and the invasive sampling procedure of CSF restricted their clinical applications.

Blood-based biomarkers have the advantages of being less invasive, more cost-effective and better feasibility[5]. Several studies have demonstrated that plasma Aβ40, Aβ42, phosphorylated tau (P-tau) and neurofilament light (NfL) are to some extent correlated with PET scan and CSF detection results[6–8]. Those biomarkers were measured on various innovative platforms, because the concentrations of plasma biomarkers are significantly lower than those in CSF in general. For each individual biomarker, the cut-point and prediction power varied in specific assays, platforms and cohorts[9,10].

Among the existing plasma biomarkers, P-tau has shown the highest predictive accuracy for Aβ positivity[11,12]. However, one single biomarker can hardly represent thorough disease status. Previous research attempted to develop algorithms combining multiple factors and showed better and more robust performance in predicting Aβ positivity of Alzheimer's disease. Those algorithms are clinically meaningful and demonstrated ideal accuracy in western cohorts ADNI and BioFINDER[11,13,14].

The increasing prevalence of AD calls urgent attention to both treatment and diagnosis[15]. Various computational models, including mathematical models, causal models, data-driven models, and personalized models, were constructed to investigate the strategies that may be utilized to target both AD treatment and diagnosis[16]. Hao and Friedman used a system of partial differential equations to simulate the efficacy of drugs that may slow the progress of the disease. The mathematical model was based on a schematic network which depends on some unconfirmed interaction between amyloid, tau, and NfL in AD[17]. Iturria-Medina et al. used a multifactorial causal model and aimed to find the causal event leading to late-onset Alzheimer's disease. The model suggested some complex interplay among multiple relevant direct interactions that ultimately cause AD. However, the causal relationship has not been confirmed at the individual level[18]. Zhu et al. constructed a subject-specific AD classifier to refine the dataset at the individual level. It could potentially achieve classification in complex and heterogeneous MRI datasets but lacked unity and interpretability at a universal level[19]. Many researches, including our study, employed data-driven models to predict AD progression[20–22]. Combinations of genes, neuropsychological scales, and biomarkers were integrated to make an accurate prediction. However, all data-driven models do not depend on any pathological assumptions. The actual results were only inferred from the underlying data. Therefore, data-driven models require a large scale of data to be reliably corroborated[23]. Under the current status of this field, the pathology of AD progression has not been clarified and confirmed. Models that rely on pathological assumptions seem fragile on where they stand. With sufficient and multi-platform data for analysis, data-driven models could directly establish connections between patient features and target results without any premise on the pathological pathways. Our study, utilizing data-driven models, attempts to find the optimal computational models to predict AD and its corresponding feature under various circumstances.

In the present study, we constructed computational models to predict brain Aβ pathology based on plasma biomarkers, *APOE* genotypes, cognitive test scores and key demographics. Our integrated models were able to accurately predict Aβ positivity, with the best performance observed in the MCI group, particularly in the aMCI subgroup. We developed other integrated models based on plasma biomarkers alone which were able to discriminate the patients in different development stages of dementia. All plasma biomarkers were found to fluctuate in participants with comorbidities or family history.

## Methods

**Study participants**. Participants were consecutively enrolled from Sixth People's Hospital, Shanghai, China. The inclusion criteria were as follows: (1) aged 40 to 85 years; (2) educated for more than one year and fully understand the neuropsychological tests; (3) consent to the blood tests, cranial MRI and 18F-florbetapir PET scan. The exclusion criteria were as follows: (1) significant systemic illness or renal and hepatic dysfunction which may interfere with the results of plasma biomarkers; (2) Individuals with a history of significant neurologic disease and psychiatric disorders; (3) other conditions which may be adversely affecting cognitive function. To evaluate the performance of our integrated model, we hypothesized that our integrated model would exceed the current standard clinical method to diagnose AD in AUC. According to clinical data, the standard AUC is defined as 0.85 and our model is expected to have an AUC of 0.90. According to statistical convention, we set a two tailored $\alpha = 0.05$ and $\beta = 0.2$. Using the formula

$$N = \frac{Z_{1-\alpha}\sqrt{V_{H0}(\widehat{AUC})} + Z_{1-\beta}\sqrt{V(\widehat{AUC_{new}}) + V(\widehat{AUC_{standard}})}}{AUC_{new} - AUC_{standard}},$$

we determined the minimum sample size required is 415. As a result, 609 participants with available data from both blood tests and 18F-florbetapir PET scan within 3 months after blood sampling were included in this study. The sample size is sufficient to reject the null hypothesis[24]. Written informed consents were obtained from all the participants or their caregivers. The ethics committee of Shanghai Jiao Tong University Affiliated Sixth People's Hospital approved this study.

**Neuropsychological assessment and diagnostic classification**. The Chinese version of the Mini-Mental State Examination (MMSE)[25], Montreal Cognitive Assessment-Basic (MoCA-BC)[26], and Addenbrooke's Cognitive Examination-III (ACE-III-CV)[27] were selected as brief cognitive screening tests. Global functional status was assessed by Activities of Daily Living (ADL) and Functional Assessment Questionnaire (FAQ)[28]. Different cognitive domains were assessed by a battery of standardized neuropsychological tests, including Auditory Verbal Learning Test (AVLT)[29] 20 min delayed free recall and AVLT recognition for memory, Boston Naming Test (BNT)[30] and Animal Verbal Fluency Test (AFT)[31] for language, Shape Trail Test Part A and B (STT-A, STT-B)[32] for executive function.

The clinical diagnosis of probable Alzheimer's disease dementia was made by experienced neurologists according to the National Institute on Aging and Alzheimer's Association (NIA-AA) criteria[33], and only those with positive results of 18F-florbetapir PET scan were classified as Alzheimer's disease dementia patients. Participants met the criteria for all-cause dementia but not classified as Alzheimer's disease dementia were classified as non-Alzheimer's disease dementia patients. Participants with cognitive symptoms but not met the diagnostic criteria for dementia were classified as MCI if they met the actuarial neuropsychological criteria put forward by Jak and Bondi[34].

The subgroup of aMCI was defined based on the impaired performance on AVLT delayed free recall and AVLT recognition. In the participants who performed essentially normal on neuropsychological tests, those with self-reported and concerned memory decline within the last 5 years and onset age over 60 years were classified as subjective cognitive decline (SCD)[35].

**Measurements of plasma Aβ42, Aβ40, T-tau, P-tau181 and NfL.** The latest Simoa technique (Single Molecule Array, by Quanterix) was used for plasma biomarker detection. All of these five biomarkers were detected using two steps reaction methods. Aβ42, Aβ40 and T-tau were detected using Neurology 3-Plex A Assay Kit (Lot 502838). For the first step, 25 μl bead, 20 μl detector and 38 μl sample were diluted with 114 μl diluent, the incubation time was 47 cadences (45 s per cadence). For the second step, 100 μl β-galactosidase-streptavidin (SBG) was added into the reaction mixture, and the incubation time was 7 cadences. 50 μl resorufin β-D-galactopyranoside (RGP) was added for the measurement. P-tau 181 was detected using P-tau 181 Assay Kit V2 (Lot 502923). For the first step, 25 μl bead, 20 μl detector and 25 μl sample were diluted with 75 μl diluent, the incubation time was 47 cadences. For the second step, 100 μl SBG was added into the reaction mixture, and the incubation time was 7 cadences. 50 μl RGP was added for the measurement. NfL was detected using NF-light Assay Kit (Lot 202700). For the first step, 25 μl bead, 20 μl detector and 38 μl sample were diluted with 114 μl diluent, the incubation time was 47 cadences. For the second step, 100 μl SBG was added to the reaction mixture, and the incubation time was 7 cadences. 50 μl RGP was added for the measurement. Twenty-four samples were tested using duplicate measurements to ensure the repeatability of our experiment based on Simoa platform, and the remaining 585 samples were tested using singlicate measurement. All the samples were analyzed on one occasion.

**Parameters calibration for model construction.** A few restrictions on all of the models were applied when constructing the decision tree models. The minimum number of observations for a split to be attempted was set to eight. The minimum number of observations in any terminal leaf node was set to four. The maximum tree depth was restricted to be the same as the number of variables in the model. All the parameters stated above were used to avoid over-fitting. The best model was first chosen based on cross-validation (CV) error rate. The model, which gives the maximum CV error rate within one standard deviation from the lowest CV error rate, was deemed as giving the best trade-off between model complexity and model fit. Then, variables were deleted in the sequence of their variable importance. The deletion process only stops until there was a significant difference in AUC to the original model based on Delong's test. The refined model further restricted the tree depths to three with all other parameters unchanged.

**Amyloid PET imaging.** The 18F-florbetapir PET scans were performed with a PET/CT system (Biograph mCT Flow PET/CT, Siemens, Erlangen, Germany) 50 min after the intravenous injection of 7.4 MBq/kg 18F-florbetapir and lasted for 20 min. PET images were reconstructed by filtered back projection algorithm with corrections for decay, normalization, dead time, photon attenuation, scatter and random coincidences. In brief, images were coregistered to the individual structural MRI and further warped into the standard Montreal Neurological Institute (MNI) stereotactic space. Standard uptake value ratios (SUVR) were calculated for the cortical regions of interest (ROIs) using cerebellar crus as a reference[36]. The mean cortical SUVR scores were calculated by weighted averaging of these ROIs. PET Image interpretation was performed by three nuclear medical physicians with specialized training according to the guidelines of visual rating[37].

**Statistical analyses.** APOE genotypes were categorized into different groups according to their Aβ risks (41): (a) ε2/ε2 or ε2/ε3; (b) ε3/ε3; (c) ε2/ε4 or ε3/ε4; (d) ε4/ε4. Categorical variables were analyzed by Chi-squared test and multi-group comparisons of continuous variables were assessed by Kruskal–Wallis test. Mann-Whitney U test was conducted to compare the differences in plasma biomarkers between the participants of amyloid PET positive and negative in each diagnostic group. Decision tree models constructed with recursive partitioning were performed. The full model, which provided excessive accuracy, utilized all available variables and restricted the tree depth to be the same as the number of variables in the model. Next, the best model was constructed by two steps fine-tuning: 1) complexity reduction by the best trade-off between model complexity and model fit according to the CV error rate; 2) variable reduction by removing as many variables as possible without significant decrease of model performance. A refined model was created by restricting the tree depths to 3, in the effort to provide simple yet concise PET positivity prediction rules for the clinicians. Receiver operating characteristic curves (ROC) were produced and AUC, sensitivity, specificity, positive predictive value (PPV), negative predictive value (NPV) and accuracy were calculated under the optimal Youden threshold to evaluate the performance of prediction models. Differences in AUCs were compared using the DeLong method[38]. Models were trained and tested through 1000 times CV to avoid over-fitting and maximize model stability. Validation in ADNI cohort was performed using the same variables from the training cohort. The DeLong's test was used to compare the performance in pairing diagnostic groups between the training cohort and ADNI cohort. A two-sided P value < 0.05 was considered statistically significant. The cut-off values for each node were z-score transformed. All data analyses were performed with R software 4.0.3.

**Reporting summary.** Further information on research design is available in the Nature Portfolio Reporting Summary linked to this article.

## Results

**Clinical characteristics of participants in the cohort.** Participants were enrolled by the process described in Methods. Participants with complete demographic information, β-amyloid PET results, clinical diagnostic results and most of the cognitive assessments and plasma biomarker measurements were included in this study. Those resulted in 609 individuals in total. Based on the diagnostic criteria mentioned in Methods, the participants were classified as cognitive normal (CN (n = 238), SCD (n = 118), MCI (n = 135, including 93 aMCI), Alzheimer's disease dementia (n = 89) and non-Alzheimer's disease dementia (n = 29). Baseline characteristics of this cohort were presented in Supplementary Data 1 and plasma biomarkers in Supplementary Table 1. The group of CN showed relatively younger ages, and the groups of MCI, Alzheimer's disease and non-Alzheimer's disease dementia had significantly lower education levels compared to cognitive normal controls. No significant difference in sex or body mass index (BMI) was found among five groups. The Alzheimer's disease group showed significantly higher frequency of APOE ε4 genotype, while no significant difference was found in the groups of SCD, MCI and non-Alzheimer's disease dementia. A detailed distribution of APOE genotypes was summarized in

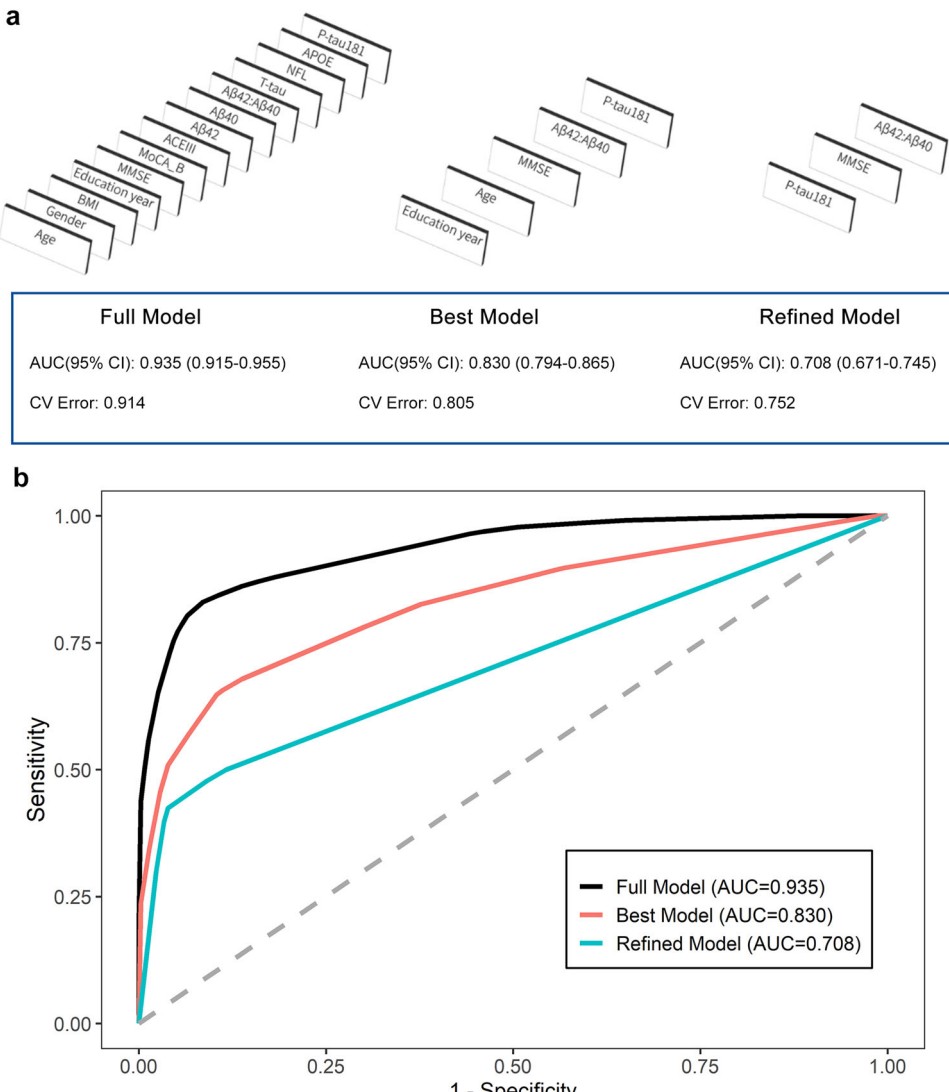

**Fig. 1 Integrated models in the whole dataset.** Model selection process and performance for predicting PET positivity in all the participants ($n = 609$).
**a** The decision tree model selection process. The full model was generated with all covariates. The best model was pruned with the best cross-validation error rate and deleted as many variables as possible while maintaining similar performance. The refined model restricted the tree depths to three.
**b** Receiver operating characteristic (ROC) curves of the different models for discriminating Aβ-PET positive versus negative.

Supplementary Table 2. In order to evaluate their cognitive and functional status, all participants completed most of the assessment scales (Methods). Participants of MCI, Alzheimer's disease and non-Alzheimer's disease dementia had worse performance in the brief cognitive assessments and standardized neuropsychological tests than CN, and participants of Alzheimer's disease and non-Alzheimer's disease dementia had worse performance in ADL and FAQ than CN. There were 46 (19%), 36 (31%) and 53 (39%) Aβ positive cases in CN, SCD, MCI groups respectively. Based on our diagnostic criteria, dementia patients with positive 18F-florbetapir PET scan results were classified as Alzheimer's disease, while the rest as non-Alzheimer's disease dementia (Methods).

**Integrated model accurately predicted Aβ positivity.** Each individual plasma biomarker had limited performance in predicting Aβ status solely (Supplementary Table 3 and Supplementary Fig. 1). Therefore we combined multiple variables including plasma biomarker measurements, *APOE* genotypes, cognitive test scores and key demographics to construct

integrated prediction models. The model selection process and AUC of all steps were illustrated in Fig. 1a. The ROC curves of the three models in all participants were shown in Fig. 1b. The detailed statistic parameters of models were shown in Supplementary Table 4. The whole process was completely data-driven, without any prior weighting or setting for the variables. First, variables were automatically selected by their importance and formed a full model. The AUC of full model is 0.94 (95% confidence interval (CI): 0.92–0.96). Although the full model has high accuracy in predicting Aβ positivity, the clinical application of it may be challenging since it requires most of the variables. Therefore we streamlined it to a best model by two steps of fine-tuning: 1) complexity reduction by the best trade-off between model complexity and model fit; 2) variable reduction by removing as many variables as possible without significant decrease in model performance. The best model yielded an AUC of 0.83 (95% CI: 0.79–0.87) while it contains only 5 variables: plasma P-tau181, plasma Aβ42/Aβ40 ratio, MMSE score, education years and age. To further simplify the prospective application in clinical practice, we finally restricted the depths of model to three to mimic the common diagnostic path. It indicates

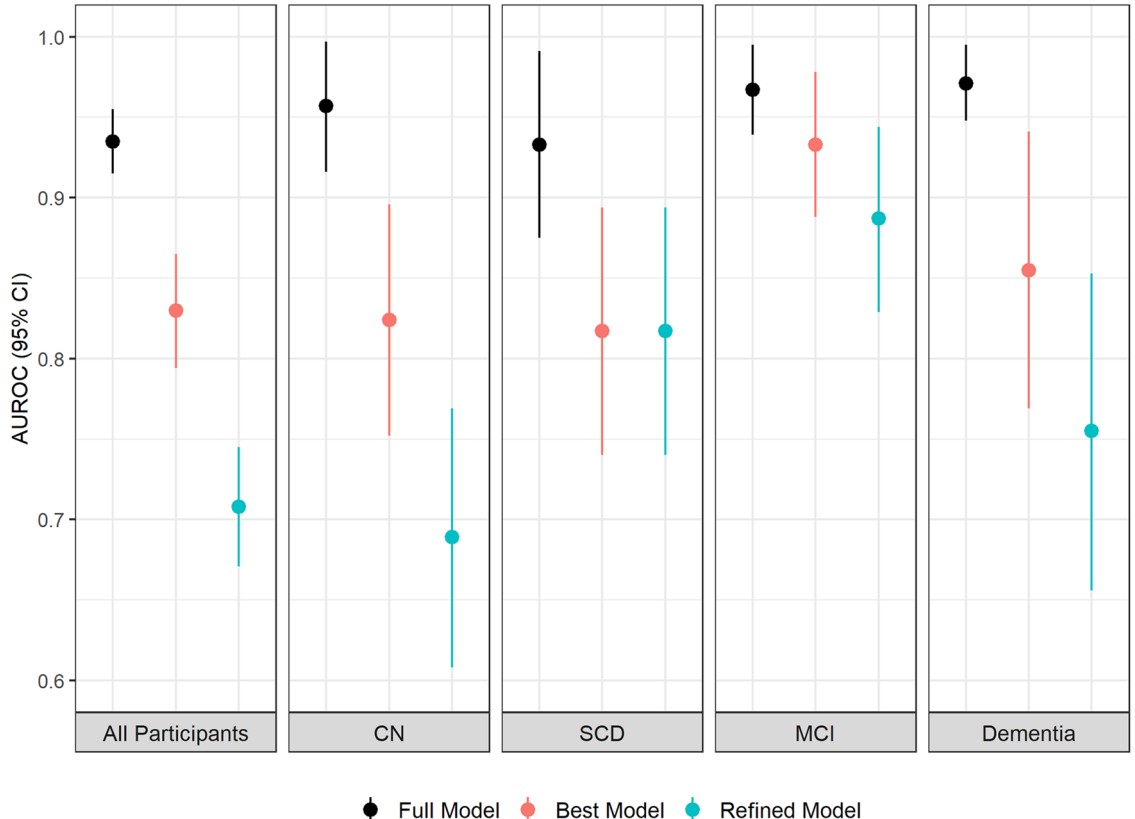

**Fig. 2 Integrated models in individual groups.** Area under the Receiver Operating Characteristic (AUROC) values and the corresponding 95% confidence intervals of the established models for predicting Aβ-PET positivity in different diagnostic groups (All participants: $n = 609$; CN: $n = 238$; SCD: $n = 118$; MCI: $n = 135$; Dementia: $n = 118$) were shown.

| Table 1 Best model performances in different dementia stages. | | | | | | | |
|---|---|---|---|---|---|---|---|
| | **AUC (95% CI)** | **Sensitivity** | **Specificity** | **PPV** | **NPV** | **Accuracy** | **CV Error** |
| General population | 0.830 (0.794–0.865) | 65.6% | 88.8% | 77.4% | 81.6% | 80.3% | 0.805 |
| CN | 0.824 (0.752–0.896) | 56.5% | 93.8% | 68.4% | 90.0% | 86.6% | 1.016 |
| SCD | 0.817 (0.740–0.894) | 86.1% | 67.1% | 53.4% | 91.7% | 72.9% | 1.025 |
| MCI | 0.933 (0.888–0.978) | 88.7% | 93.9% | 90.4% | 92.8% | 91.9% | 0.491 |
| Dementia | 0.855 (0.769–0.941) | 91.0% | 72.4% | 91.0% | 72.4% | 86.4% | 1.108 |

*AUC area under the ROC curve, PPV positive predictive value, NPV negative predictive value, 95% CI 95 percent confidence interval, CV Error cross-validation error, CN cognitive normal, SCD subjective cognitive decline, MCI mild cognitive impairment.*

that the users can get the results by no more than 3 steps inter-pretations. The refined model resulted in an AUC of 0.71 (95% CI: 0.67–0.75). The details of modeling process is described in Methods.

**The prediction of Aβ positivity in MCI subgroup had the highest accuracy.** The same procedures were carried out to construct the prediction models in each diagnostic group independently. The performance of the best models in all individual groups was demonstrated in Fig. 2 and Table 1. The top pre-diction was in MCI group with an AUC 0.93 (95% CI: 0.89–0.98). The AUCs of CN, SCD and dementia groups were 0.82 (95% CI: 0.75–0.90), 0.82 (95% CI: 0.74–0.89) and 0.86 (95% CI: 0.77–0.94), respectively. The AUC values and the corresponding

95% confidence intervals of prediction in four groups were shown in Fig. 2. The detailed statistic parameters of models in MCI subgroup were shown in Supplementary Table 4.

**The Aβ positivity in aMCI subgroup can be predicted by plasma biomarkers alone.** Afterwards, we dug into the subtypes of MCI patients. In the subgroup of aMCI, which is memory-specific and recognized as the precursor to Alzheimer's disease, the integrated models revealed superior performance. The AUCs of full model, best model and refined model are 0.97 (95% CI: 0.94–1.00), 0.96 (95% CI: 0.92–0.99) and 0.89 (95% CI: 0.84–0.95). The detailed statistical parameters of models in aMCI subgroup were shown in Supplementary Table 4. Remarkably, the variables in the automatically data-driven selected refined model

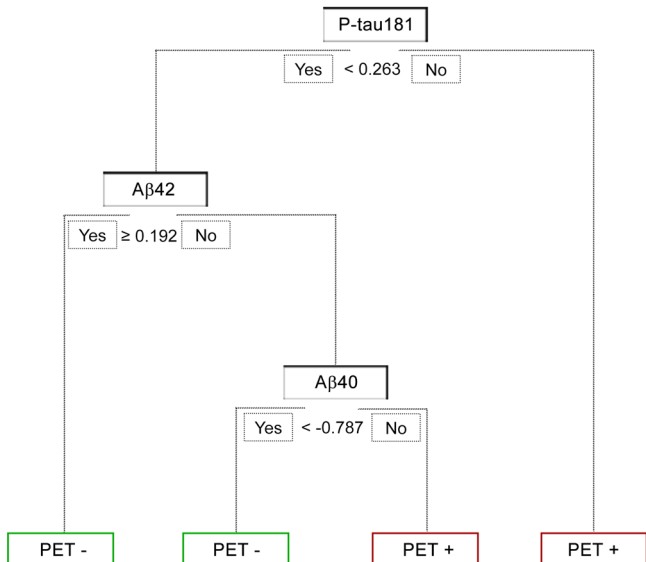

**Fig. 3 Tree view of aMCI refined model. Decision Tree diagram of the refined model in the aMCI group (n = 93).** The tree started from the root node. If the value met the current condition, the workflow would go to the left node; otherwise, the right node. The green nodes indicated PET negative and the red nodes indicated PET positive. Note that all values were transformed to z-score prior to fitting the model. Cut-off values were shown in z-score values.

were plasma P-tau181, Aβ40 and Aβ42, indicating that the Aβ status of PET-CT can be predicted with reasonable confidence by only measuring plasma biomarkers. The refined model with detailed paths and parameters were shown in Fig. 3. This concise and straightforward model provided potential possibility for clinical utility broadly.

**The integrated models were validated in the ADNI cohort.** To investigate the robustness of our integrated models, we validated them in Alzheimer's Disease Neuroimaging Initiative (ADNI) cohort. To get an impartial comparison, we tried to assemble the data which have identical variables as in our dataset. As the result, 284 cases with available data of demographics (age, sex and education years), *APOE* genotypes, plasma biomarkers (Aβ40, Aβ42, P-tau181, T-tau and NfL), brief cognitive test (MMSE) and the results of Aβ-PET were collected, including 97 of CN, 124 of MCI and 63 of Alzheimer's disease patients. SCD group in ADNI was excluded because we didn't find complete plasma biomarker data for those. The demographic summary of validation dataset was shown in Supplementary Data 1. There was 8 years difference in the average ages of two cohorts (77 years old in ADNI cohort and 65 years old in our study). The proportion of *APOE* ε4 carriers are also higher in ADNI cohort (30.9–69.8%) than in our study (14.4–52.8%), which consists with earlier finding that the frequency of *APOE* ε4 genotype is lower in Asian than western population[39]. For the cognitive test in ADNI dataset, only MMSE was available and therefore involved in the validation process. BMI was not involved in the procedure of model selection because it was only available in a small population in ADNI. We compared the levels of Aβ40, Aβ42, NfL, P-tau181, and T-tau between ADNI dataset and our dataset. ADNI cohort had an overall higher distribution than the Chinese cohort based on the raw value after measurement (Supplementary Table 1 and Supplementary Fig. 2). These differences may be caused by the experimental kit and the ethnic features between ADNI and our study. In order to have unified values as an input to the model, all

values were z-score transformed within their own dataset. After the transformation, only Aβ40, and Aβ42 have statistical difference between the ADNI and the Chinese cohort in NC and MCI group (Supplementary Fig. 3).

The same variables as previously described were input to construct models in all the collected cases from ADNI. No non-Alzheimer's disease dementia data with all five biomarkers were found in ADNI and therefore the analyses were performed in the whole dataset, CN and MCI subgroup separately. As shown in Fig. 4 and Supplementary Table 5, the full model, best model and refined model in ADNI returned similar accuracy to this study (AUC = 0.96 (95%CI: 0.93–0.98), 0.88 (95%CI: 0.84–0.92) and 0.75 (95%CI: 0.70–0.81) in the whole dataset; AUC = 0.91 (95% CI: 0.85–0.98), 0.86 (95%CI: 0.78–0.93) and 0.71 (95%CI: 0.63–0.79) in the subgroup of CN; AUC = 0.95 (95%CI: 0.91–0.99), 0.93 (95%CI: 0.89–0.97) and 0.87 (95%CI: 0.81–0.93) in the subgroup of MCI). The AUCs of the whole dataset in ADNI were higher than in our cohort. It is probably because the ADNI dataset in this study did not contain SCD group, as the models had the lowest prediction AUCs in SCD group in previous section (Fig. 2 and Table 1). This assumption was proven by matching our cohort population by removing SCD and non-Alzheimer's disease patients from the dataset. The resulting AUCs were 0.95 (95%CI: 0.92–0.97), 0.87 (95%CI: 0.84–0.91) and 0.79 (95%CI: 0.75–0.83), which does not differ significantly from that obtained from ADNI cohort (*p*-value > 0.05). Furthermore, the best models in the whole dataset and MCI subgroup in ADNI cohort both had exactly identical variables (MMSE, plasma P-tau181, plasma Aβ42/Aβ40 ratio, education years and age for the whole dataset; *APOE* genotype, plasma P-tau181, plasma Aβ42/Aβ40 ratio and plasma Aβ40 for the MCI subgroup) as in our cohort. Those findings suggested that the integrated models established in our Chinese cohort can be effectively applied in another independent cohort, even with different ethnic backgrounds.

**Integrated model of plasma biomarkers alone for distinguishing the patients in different stages.** In the context of primary care, such as community-based large-scale screening and physical examination, the cognitive tests and demographic information are sometimes missing/biased due to the lacking of experienced neurologists. Therefore we explored the opportunity to predict disease status by using only plasma biomarkers. The results of ROC analysis was shown in Fig. 5. AUCs with their 95% confidence intervals, sensitivities, specificities, PPV, NPV were shown in Supplementary Table 6. Alzheimer's disease group can be discriminated very well from CN, SCD and MCI (AUCs are 0.93 (95% CI: 0.90–0.96), 0.91 (95% CI: 0.87–0.96) and 0.92 (95% CI: 0.88–0.96), respectively), suggesting that the Alzheimer's disease patients can be identified nicely by using just blood biomarkers testing. The AUCs in other groups were not as good as Alzheimer's disease: 0.75 (95% CI: 0.69–0.81) in SCD vs. MCI, 0.74 (95% CI: 0.68–0.79) in CN vs. SCD and 0.70 (95% CI: 0.65–0.74) in CN vs. MCI. Those observations indicated that plasma biomarkers are probably not completely sufficient to specify early progress stages of dementia.

**The impacts of comorbidities and family history.** Part of the participants in our study had records of chronic disease and family history. To investigate the influence of disease and family history on the plasma biomarkers and prediction models, we analyzed the distribution of plasma biomarker values and the performance of integrated models in disease and family history-related participants. Diseases/family histories with at least 40 patients in our study were involved, including respiratory system

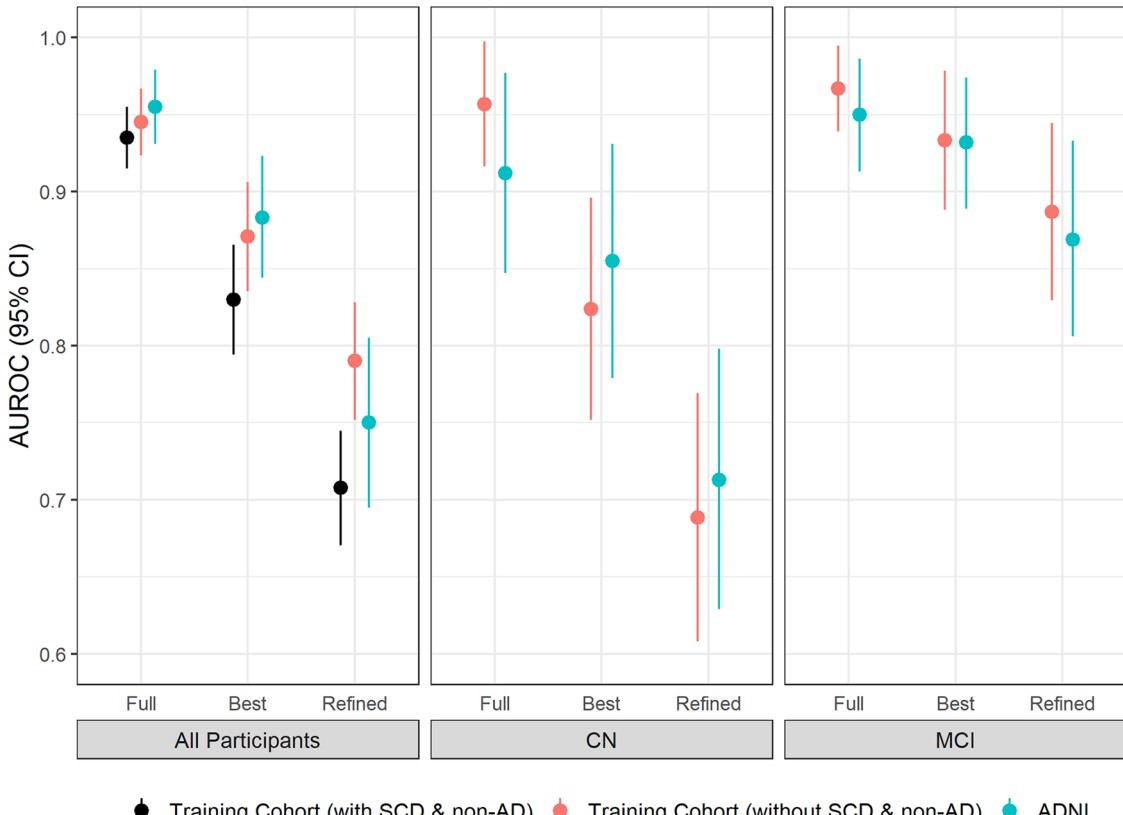

**Fig. 4 Model validation in ADNI cohort.** AUROC values and the corresponding 95% confidence intervals of the established model for predicting Aβ-PET positivity in different diagnostic groups and in different cohorts. The n number of each group is showed as follows: All participants: Training with SCD & non-AD: n = 609; Training without SCD & non-AD: n = 462; ADNI: n = 284. CN: Training: n = 238; ADNI: n = 97. MCI: Training: n = 135; ADNI: n = 124.

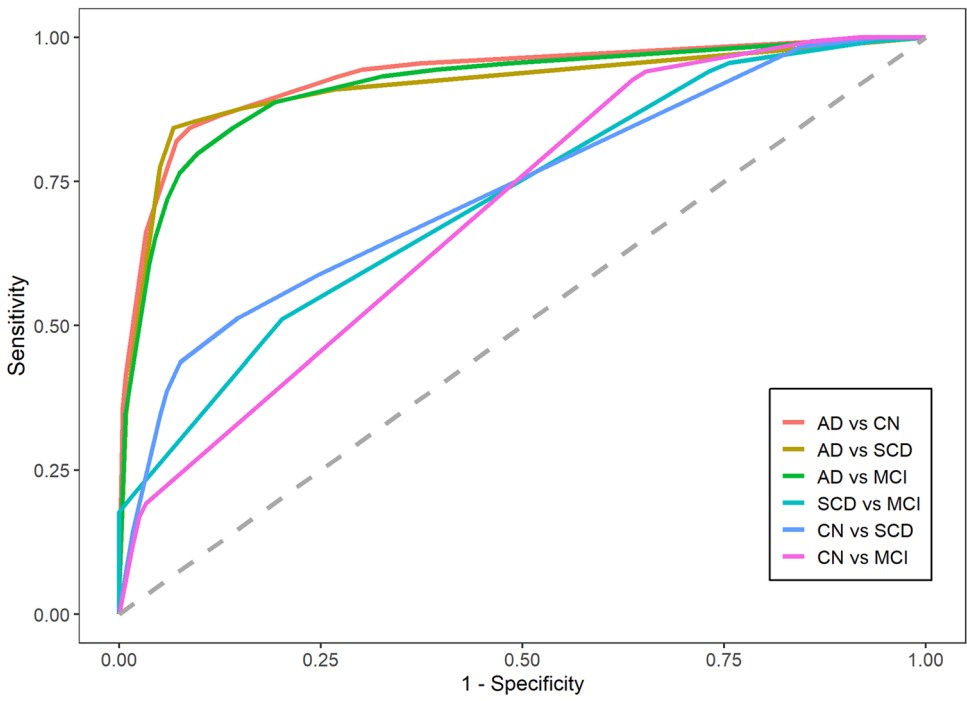

**Fig. 5 Prediction of disease stages by plasma biomarkers alone.** ROC plots for showing the efficiency of the integrated model for distinguishing the patient with different dementia statuses. (CN: n = 238; SCD: n = 118; MCI: n = 135; Alzheimer's disease: n = 89).

disease, coronary heart disease, hypertension, hyperlipemia, diabetes, general anesthesia history and Alzheimer's disease family history. The analysis details were demonstrated in Methods. The plasma biomarker values with/without disease/history were shown in Table 2. All five biomarkers (Aβ40, Aβ42, T-au, P-tau181 and NfL) had higher levels in hypertension patients than participants without hypertension. Aβ40 levels were elevated in hyperlipemia patients. Aβ40 and NfL levels were elevated in diabetes patients. NfL levels were slightly decreased in participants with Alzheimer's disease family history.

Next we investigated whether the changes of plasma biomarker levels influenced the prediction of integrated models. For the disease/family history with plasma biomarker value changes (hypertension, hyperlipemia, diabetes and Alzheimer's disease family history), the full model, best model and refined model were applied on participants with and without the disease/family history independently. As shown in Supplementary Fig. 4 and Supplementary Table 7, the models performed better in participants without disease/family history except for the full model for Alzheimer's disease family history. The important values for the variables in all of the models were listed in Supplementary Tables 8–25.

## Discussion

In this study, we integrated plasma biomarkers measurements, *APOE* genotypes, cognitive test scores and key demographics and developed computational models to predict brain Aβ pathology in a large Chinese cohort. We had four major findings. First, the integrated models can predict Aβ positivity accurately, and the results were validated in ADNI cohort. Second, the models performed best in MCI group. Especially in aMCI subgroup, the model which only consisted of plasma biomarkers can accurately predict Aβ positivity with an AUC of 0.89. Third, we developed other integrated models based on plasma biomarkers alone which can discriminate the patients in different development stages of dementia. Last, all plasma biomarkers were found to fluctuate in participants with comorbidities or family history. In general, the model performance was better in participants without comorbidities or family histories.

To our knowledge, the present study is the largest Chinese cohort which included CN, SCD, MCI and Alzheimer's disease participants with comprehensive clinical diagnosis, cognitive assessments, plasma biomarker measurements and Aβ PET results. According to our inclusion criteria, the participants in this study were relatively young (average age: 65 years) comparing to ADNI and BioFinder cohorts[11,40,41]. Previous studies revealed that Aβ began to accumulate up to 15–20 years prior to clinical symptoms of Alzheimer's disease[42,43]. Therefore, it is very important to understand the early signal of disease progression. Most of the previous research were based on western population[14,42,44]. However, it is known that some essential features are discrepant in different populations, for example, the frequency of *APOE* ε4 carriers, lifestyle, etc[39,45]. Therefore it is critical to examine if the observations based on western population can be generalized to Chinese population.

The correlations between plasma biomarkers and Aβ pathology or disease status have been investigated in several studies[13,46–48]. However, few studies have reported cutpoints of those biomarkers for disease diagnosis, because cutpoints can differ based on the assays, experimental platforms and specific applications. Furthermore, few studies included all five stages of dementia status. The diagnostic power of the biomarkers also varied in different studies[49,50]. In clinical practice, it happens that the biomarkers change to contrary directions, which raises difficulties for clinicians to draw conclusions[5,51]. Therefore we recognized that the

**Table 2 Plasma biomarker levels in patients with different disease history.**

| Disease History | Respiratory Disease | | CHD | | Hypertension | | Hyperlipemia | | Diabetes | | Family History | | AD | |
| --- | --- | --- | --- | --- | --- | --- | --- | --- | --- | --- | --- | --- | --- | --- |
| | Without | With | Without | With | Without | With | Without | With | Without | With | | | Without | With |
| N | N = 506 (89.40%) | N = 60 (10.60%) | N = 523 (92.40%) | N = 43 (7.60%) | N = 363 (64.13%) | N = 203 (35.87%) | N = 465 (82.16%) | N = 101 (17.84%) | N = 477 (84.28%) | N = 89 (15.72%) | N | N = 264 (69.11%) | N = 118 (30.89%) |
| Aβ40 (pg/mL)[a] | 196.1 (61.7) | 192.2 (63.0) | 195.2 (60.9) | 201.2 (72.3) | 191.0 (60.2) | 203.9 (63.9)** | 193.3 (62.4) | 206.8 (57.8)* | 192.3 (62.1) | 213.5 (57.1)** | Aβ40 (pg/mL)[a] | 199.7 (69.0) | 192.5 (53.7) |
| Aβ42 (pg/mL)[a] | 9.9 (3.5) | 10.0 (3.4) | 9.9 (3.5) | 9.3 (4.0) | 9.7 (3.4) | 10.2 (3.6)* | 9.8 (3.5) | 10.3 (3.6) | 9.7 (3.5) | 10.7 (3.5) | Aβ42 (pg/mL)[a] | 10.2 (3.8) | 10.2 (3.3) |
| T-tau (pg/mL)[a] | 2.4 (1.1) | 2.6 (2.2) | 2.5 (1.3) | 2.3 (0.8) | 2.4 (1.3) | 2.5 (1.0)* | 2.4 (1.3) | 2.5 (1.0) | 2.4 (1.3) | 2.5 (1.1) | T-tau (pg/mL)[a] | 2.6 (1.4) | 2.4 (1.2) |
| P-tau181 (pg/mL)[a] | 2.5 (3.5) | 2.2 (1.3) | 2.5 (3.4) | 2.1 (1.1) | 2.5 (4.0) | 2.4 (1.3)* | 2.5 (3.6) | 2.3 (1.1) | 2.5 (3.6) | 2.4 (1.5) | P-tau181 (pg/mL)[a] | 2.7 (4.7) | 2.2 (1.4) |
| NfL (pg/mL)[a] | 17.6 (14.8) | 15.4 (7.0) | 17.4 (14.4) | 16.8 (12.2) | 16.3 (12.8) | 19.2 (16.2)* | 17.4 (15.2) | 17.0 (8.4) | 16.7 (13.0) | 20.5 (19.1)* | NfL (pg/mL)[a] | 18.5 (14.5) | 16.0 (15.1)* |
| Aβ42/Aβ40 ratio[a] | 0.0519 (0.016) | 0.0544 (0.016) | 0.0526 (0.016) | 0.0470 (0.015) | 0.0522 (0.015) | 0.0521 (0.018) | 0.0524 (0.016) | 0.0512 (0.015) | 0.0525 (0.017) | 0.0506 (0.011) | Aβ42/Aβ40 ratio[a] | 0.0537 (0.018) | 0.0541 (0.014) |

Data were shown as mean ± (s.d.). P values were tested by Mann-Whitney Test. *p-value < 0.05; **p-value < 0.01; ***p-value < 0.001. CHD Chronic Heart Disease.

most possible solution is to combine the disease-related variables and generate data-driven, non-biased computational models in each stage, which will dilute the contribution/noise of single factor and represent the full-scale picture. In present study, we established integrated models with three levels of accuracies and complexities in every stage, which can serve different application purposes. The full model has the highest accuracy, but it also requires complete input variables. The best model was optimized to contain only critical variables yet remaining satisfactory performance. By sacrificing certain accuracy, the refined model is very easy-to-use since it only has three steps judgements. The users can select appropriate models based on the testing conditions and clinical scenarios.

MCI is the critical stage for Alzheimer's disease progression. It was reported that about 15% of MCI patients older than 65 converted to Alzheimer's disease after two years of follow-up[1]. Most anti-Aβ drugs showed the best efficacy on Aβ-positive MCI patients in clinical trials and real-world studies[52]. Therefore verified Aβ status in MCI patients is very important for recruiting participants and future treatment. Our integrated models had the best performance in MCI group (AUC = 0.97 for the full model). Moreover, the refined model for aMCI subgroup containing only plasma P-tau181, Aβ40 and Aβ42 has an AUC of 0.89. Those high accuracies supported that our models can be widely used in clinical trial, screening and primary care.

Assessment of disease stages is crucial for determining plans of treatment and therapeutic intervention[53]. The cognitive tests are standard tools to classify disease stages and the clinical diagnoses are made by experienced neurologists. The results of cognitive tests may be affected by some non-objective factors, such as education levels, the health condition and mood of examinees and so on[54]. Therefore we tried to generate a prediction model for disease stages by using only plasma biomarkers. Alzheimer's disease patients can be clearly identified from other groups. But CN, SCD and MCI groups did not show specific patterns in this model.

Blood biomarkers are regulated by multiple metabolic pathways and circulating environments. It is known that the levels of Alzheimer's disease plasma biomarkers are affected by several factors including age, sex and comorbidities[55]. Our study also showed that the levels of Alzheimer's disease plasma biomarkers changed in patients with other comorbidities or family histories. Especially in patients with hypertension, the values of all five biomarkers increased. Furthermore, the integrated models had higher accuracy in participants without comorbidities or family histories, which suggests that clinicians should pay attention and take those factors into consideration when using the models. As a limitation, although the sample size was the largest in China to our knowledge, it was smaller than some American and European studies such as ADNI and BioFINDER.

Recently, several studies reported computational models to predict the diagnostic effectiveness of AD using supportive vector machine, logistic regression, Bayes classifier, random forest, and decision tree algorithms[13,48,54]. In our study, we selected decision tree models for four reasons. First, comparing to other machine learning methods, a decision tree is easy to understand and interpret. As a white-box model, the detailed structure and Boolean logic of a decision tree can be clearly visualized. Secondly, training a decision tree model requires much fewer data than other machine learning models, which usually require thousands of entries to avoid over-fitting. Thirdly and most importantly, in our case, decision tree models outperform other models in accuracy and stability. It provides the perfect balance within the accuracy and interpretability trade-off. Lastly, decision tree models are also not susceptible to missing values. A small portion of participants in our study didn't complete all neuropsychological assessment examinations. Machine learning models such as supportive vector machine have difficulties handling those missing values.

We present an innovative approach to accurately predict brain Aβ pathology by integrated models combining plasma biomarkers, *APOE* genotypes, cognitive test scores and key demographics. The models were established in a Chinese cohort and were validated in an independent western cohort (ADNI). The prediction accuracy of models in MCI population was the highest (AUC = 0.97). This non-invasive and easily accessible method may have many applications which can help early identification of potential Alzheimer's disease patients, enrollment of Alzheimer's disease drug clinical trials and assessment of Alzheimer's disease treatments.

## Data availability

The ADNI dataset used in this study were stored at https://ida.loni.usc.edu. The other raw participant data were restricted to protect the privacy of individuals, so the data are not publicly available, and these data are available from the authors upon reasonable request and with permission from Shanghai Jiao Tong University Affiliated Sixth People's Hospital.

## Code availability

The source code that support the findings of this study is available on GitHub at https://github.com/WXDX-DA/AbetaPrediction and on Zenodo at https://zenodo.org/record/7844160#.ZEd7BM5ByUl[56].

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

## Acknowledgements

We would like to thank Dr. Ronald C. Petersen and Dr. Jonathan Graff-Radford from Mayo Clinic for their insightful discussion and intellectual content. This work was supported by the National Natural Science Foundation of China (82171198), Shanghai Municipal Science and Technology Major Project (No. 2018SHZDZX01), ZJLab the Guangdong Provincial Key S&T Program (2018B030336001) and Shanghai Pujiang Program (21PJ1423100).

## Author contributions

Contributors Z.F. and Q.G. conceptualized and designed the study. Z.F., Q.G., X.C., F.P., F.X., J.Z. and Y.Z. drafted the initial manuscript and reviewed and revised the manuscript. Z.F., X.C., J.D., D.Y. and J.Z. generated the computational models and did the statistical analysis. F.P., Y.H., Y.W. and Q.G. completed patient recruitment, cognitive assessment and *APOE* genotype measurement. Y.G. and F.X. completed PET scan and the data analysis of the PET. All authors approved the final manuscript as submitted and agree to be accountable for all aspects of the work.

## Competing interests

The authors declare no competing interests.
