## [Peer Review File · Communications Medicine]

Reviewers' comments:

Reviewer #1 (Remarks to the Author):

In the manuscript entitled "Integrated algorithm combining plasma biomarkers and cognitive assessments accurately predicts brain β -amyloid pathology: a large Chinese cohort study" by Pan et al., the authors have collected multiple variables from a large Chinese cohort and utilized computational models in predicting A β positivity. They also validated the models with additional ADNI data. The study provided an innovative strategy for the assessment of A β pathology through plasma biomarkers. Overall, the data is presented in a logic order, and it is a good paper making the main points.

I have some minor concerns regarding the manuscript.

1. Page 5, line 131. The authors adopted the decision tree for the construction of prediction models, and I wonder why they select this method? Why don't they apply other supervised machine-learning methods, such as supportive vector machine, Bayes classifier, or random forest? The reasons should be discussed.
2. Page 7, line 176. Please provide the Gini value or Importance value for the variables in each model. Especially in the "Full model", the Gini value or Importance value would assist readers to better understand how to select the proper variables and refine the models.
3. Page 10, line 264. With the five plasma biomarkers, the authors constructed the computational models between every two disease statuses (Fig. 5). I am wondering if it is feasible to construct a single model to distinguish all patients at different statuses, which would improve value for clinical diagnosis of the study.
4. Page 12, line 326: " The correlation between plasma biomarkers and A β pathology or disease status have been investigated in several studies". The authors should provide related references.

Reviewer #2 (Remarks to the Author):

This is an important and interesting study that confirms the diagnostic performance of plasma biomarkers for Alzheimer's pathologies together with APOE genotype, cognitive testing, and basic demographics data in an Asian cohort. The results were replicated in ADNI. The manuscript is well-written and clear. My only criticism is confined to Methods. Please describe in greater detail exactly how the biomarker measurements were performed. Were all samples analysed on one occasion? Were singlicate or duplicate measurements used? What was the CV for the respective assays?

Reviewer #3 (Remarks to the Author):

This paper develops a computational model to predict brain A β positivity and validates the proposed method on ADNI dataset. I would like the authors to address my following comments before further considering this publication:

1. The paper develops a computational model but lacks a discussion of the state-of-art in math/computational model of AD. By simply googling this topic, many mathematical models, causal models, data-driven models, and personalized models with optimal treatments have been developed. The authors should summarize the work in the introduction and distinguish what's the difference in their work.
2. How to calibrate the proposed model is unclear. The authors may write more details on the parameter estimation.
3. What's the difference between ADNI dataset and the Chinese cohort? Can this modeling approach capture this difference?

Point-by-Point Response to Reviewers' comments

Communications Medicine manuscript ID: COMMSMED-22-0360.

Authors: Fengfeng Pan *et al.*

General response: We kindly thank editor and reviewers for their constructive suggestions. In response to editor and reviewers' comments, we have followed their suggestions to revise the manuscript carefully. We believe that this version of the manuscript is largely improved.

Point-by-point response to the comments of reviewer #1:

In the manuscript entitled "Integrated algorithm combining plasma biomarkers and cognitive assessments accurately predicts brain β -amyloid pathology: a large Chinese cohort study" by Pan et al., the authors have collected multiple variables from a large Chinese cohort and utilized computational models in predicting A β positivity. They also validated the models with additional ADNI data. The study provided an innovative strategy for the assessment of A β pathology through plasma biomarkers. Overall, the data is presented in a logic order, and it is a good paper making the main points.

I have some minor concerns regarding the manuscript.

1. Page 5, line 131. The authors adopted the decision tree for the construction of prediction models, and I wonder why they select this method? Why don't they apply other supervised machine-learning methods, such as supportive vector machine, Bayes classifier, or random forest? The reasons should be discussed.

Response: Thanks for your questions. Recently, several studies reported computational models to predict the diagnostic effectiveness of AD using supportive vector machine, logistic regression, Bayes classifier, random forest, and decision tree algorithms¹⁻³. In our study, we selected decision tree models for four reasons. First, comparing to other machine learning methods, decision tree is easy to understand and interpret. As a white-box model, the detailed structure and Boolean logics of decision tree can be clearly visualized. Secondly, training a decision tree model requires much less data than other machine learning models, which usually requires thousands of entries to avoid over-fitting. Thirdly and most importantly, in our case, decision tree models outperform other

models in accuracy and stability. It provides the perfect balance within the accuracy and interpretability trade-off. Lastly, decision tree models are also not susceptible to missing values. A small portion of participants in our study didn't complete all neuropsychological assessment examinations. Machine learning models such as supportive vector machine have difficulties to handle those missing values. Thanks for your valuable suggestion, we have added one paragraph in the Discussion section to address this point in the revised manuscript (pages 15-16, lines 435-448).

2. Page 7, line 176. Please provide the Gini value or Importance value for the variables in each model. Especially in the "Full model", the Gini value or Importance value would assist readers to better understand how to select the proper variables and refine the models.

Response: Thanks for your constructive suggestion. We have added the Importance value for the variables in all of the models in the revised manuscript (page 13 lines 363-364 and **Supplementary Table 8-25**, marked in green).

3. Page 10, line 264. With the five plasma biomarkers, the authors constructed the computational models between every two disease statuses (Fig. 5). I am wondering if it is feasible to construct a single model to distinguish all patients at different statuses, which would improve value for clinical diagnosis of the study.

Response: Thanks for your question. Actually we did have constructed a single model to distinguish all patients at different statuses. However, the performance of the single model was not ideal. The multi-class area under the ROC curves (AUC) was 0.744 and the accuracy was 0.729. The unsatisfactory prediction accuracy could be resulted from several reason. First, the number of samples is unbalanced between each status which creates significant difficulty in a multi-classification tree. Further, the features of AD group significantly differs from other groups, while the differences among NC, SCD and MCI groups were relatively mild comparing to their differences with AD group. Therefore we decided to construct computational models between every two disease statuses instead of one single model to distinguish all patients at different statuses.

4. Page 12, line 326: "The correlation between plasma biomarkers and A β pathology or

disease status have been investigated in several studies". The authors should provide related references.

Response: Thanks for your suggestion. We have provided the related references to the revised manuscripts (page 14 line 391, page 18 lines 520-523, and page 22 lines 623-632, marked in green).

Point-by-point response to the comments of reviewer #2:

This is an important and interesting study that confirms the diagnostic performance of plasma biomarkers for Alzheimer's pathologies together with APOE genotype, cognitive testing, and basic demographics data in an Asian cohort. The results were replicated in ADNI. The manuscript is well-written and clear. My only criticism is confined to Methods. Please describe in greater detail exactly how the biomarker measurements were performed. Were all samples analysed on one occasion? Were singlicate or duplicate measurements used? What was the CV for the respective assays?

Response: Thanks for your supportive comments. We have added one paragraph to the Method section to describe in greater detail exactly how the biomarker measurements were performed in the revised manuscript (pages 5-6 lines 136-155, marked in green). Yes, all the samples were analyzed on one occasion, and we have added this information to the revised manuscripts (page 6, lines 155, marked in green). Twenty-four samples were tested using duplicate measurements to ensure the repeatability of our experiment based on Simoa platform. Coefficients of variation (CV) were shown in the following figure. The remaining samples were detected using singlicate measurement. We have added the information in the revised manuscript (page 6 lines 152-155, marked in green).

Figure legend. CV values of Aβ40, Aβ42, T-tau, P-tau181 and NfL of 24 samples for ensuring the reproducibility of our experiments based on Simoa platform.

Point-by-point response to the comments of reviewer #3:

This paper develops a computational model to predict brain Aβ positivity and validates the proposed method on ADNI dataset. I would like the authors to address my following comments before further considering this publication:

1. The paper develops a computational model but lacks a discussion of the state-of-art in math/computational model of AD. By simply googling this topic, many mathematical models, causal models, data-driven models, and personalized models with optimal treatments have been developed. The authors should summarize the work in the introduction and distinguish what's the difference in their work.

Response: Thanks for your constructive suggestion. We have added a paragraph in Introduction section to summarize the work and distinguish what's the difference in these works in the revised manuscript (page 3 lines 59-84 and pages 18-19 lines 527-553).

2. How to calibrate the proposed model is unclear. The authors may write more details on the parameter estimation.

Response: Thanks for your suggestion. A few restrictions on all of the models were applied when constructing the decision tree models. The minimum number of

observations for a split to be attempted was set to eight. The minimum number of observations in any terminal leaf node was set to four. The maximum tree depth was restricted to be the same as the number of variables in the model. All the parameters stated above were used to avoid over-fitting. The best model was first chosen based on cross-validation (CV) error rate. The model, which gives the maximum CV error rate within one standard deviation from the lowest CV error rate, was deemed as giving the best trade-off between model complexity and model fit. Then, variables were deleted in sequence of their variable importance. The deletion process only stops until there was a significant difference in AUC to the original model based on DeLong's test. The refined model further restricted the tree depths to three with all other parameters unchanged. We have added more details on the parameters estimation during the calibration of the proposed model (page 6 lines 156-168, marked in green).

3. What's the difference between ADNI dataset and the Chinese cohort? Can this modeling approach capture this difference?

Response: Thanks for your questions. The Alzheimer's Disease Neuroimaging Initiative (ADNI) is a longitudinal multicenter study designed to develop clinical, imaging, genetic, and biochemical biomarkers for the early detection and tracking of Alzheimer's disease (AD). Since its launch more than a decade ago, the landmark public-private partnership has made major contributions to AD research, enabling the sharing of data between researchers around the world. However, the ADNI dataset only covers western population, mostly American. Few studies on AD research are based on Chinese population. To our knowledge, our study is the largest Chinese cohort which included cognitive normal (CN), subjective cognitive decline (SCD), mild cognitive impairment (MCI) and AD participants with comprehensive clinical diagnosis, cognitive assessments, plasma biomarker measurements and A β PET results. One of the aims of our study is to examine if the observations based on western population can be generalized to Chinese population.

Consistent with our training cohort, we selected patients who had all five biomarkers and 18F-florbetapir PET scan in the ADNI dataset. As a result, 284 cases with available

data of demographics (age, sex and education years), *APOE* genotypes, plasma biomarkers (A β 40, A β 42, P-tau181, T-tau and NfL), brief cognitive test (MMSE) and the results of A β -PET were collected, including 97 of CN, 124 of MCI and 63 of Alzheimer's disease patients. SCD group in ADNI was excluded because we didn't find complete plasma biomarker data for those (**Table 1**).

We have added the comparison of the levels of A β 40, A β 42, NfL, P-tau181, and T-tau between ADNI dataset and our dataset. ADNI cohort has an overall higher distribution than the Chinese cohort based on the raw value after measurement (**Supplementary Figure 3**, marked in green). These differences may be caused by the experimental kit or the ethnic feature between ADNI and our study. In order to have unified values as an input to the model, all values were z-score transformed within their own dataset. After the transformation, only A β 40, and A β 42 have statistical difference between the ADNI and the Chinese cohort in NC and MCI group (**Supplementary Figure 4**, marked in green). We have added these comparison results to the revised manuscript (pages 10-11 lines 294-302 and **Supplementary Figures 3-4**, marked in green)

Our model captured similar performances in our dataset and the ADNI dataset. No non-Alzheimer's disease dementia data with all five biomarkers were found in ADNI and therefore the analyses were performed in the whole dataset, CN and MCI subgroup separately. As shown in **Figure 4** and **Supplementary Table 5**, the full model, best model and refined model in the ADNI returned similar accuracy to our study (AUC=0.96 (95%CI: 0.93-0.98), 0.88 (95%CI: 0.84-0.92) and 0.75 (95%CI: 0.70-0.81) in the whole dataset; AUC=0.91 (95%CI: 0.85-0.98), 0.86 (95%CI: 0.78-0.93) and 0.71 (95%CI: 0.63-0.79) in the subgroup of CN; AUC=0.95 (95%CI: 0.91-0.99), 0.93 (95%CI: 0.89-0.97) and 0.87 (95%CI: 0.81-0.93) in the subgroup of MCI). Furthermore, the best models in the whole dataset and MCI subgroup in ADNI cohort both had exactly identical variables (MMSE, plasma P-tau181, plasma A β 42/A β 40 ratio, education years and age for the whole dataset; *APOE* genotype, plasma P-tau181, plasma A β 42/A β 40 ratio and plasma A β 40 for the MCI subgroup) as in our cohort.

Those findings suggested that the integrated models established in our Chinese cohort can be effectively applied in another independent cohort, even with different ethnic features.

Besides, some differences also have been found between the ADNI dataset and our dataset. The AUCs of the whole dataset in ADNI were higher than in our cohort. It is probably because the ADNI dataset in this study did not contain SCD group, as the models had the lowest prediction AUCs in SCD group in previous section (**Figure 2** and **Table 2**). The proportion of *APOE* $\epsilon 4$ carriers is also higher in the ADNI cohort (30.9%-69.8%) than in our study (14.4%-52.8%). The result is consistent with earlier finding that the frequency of *APOE* $\epsilon 4$ genotype is lower in Asian than western population⁴.

Reference

- 1 Palmqvist, S. *et al.* Accurate risk estimation of beta-amyloid positivity to identify prodromal Alzheimer's disease: Cross-validation study of practical algorithms. *Alzheimers Dement* **15**, 194-204, doi:10.1016/j.jalz.2018.08.014 (2019).
- 2 Benedet, A. L. *et al.* The accuracy and robustness of plasma biomarker models for amyloid PET positivity. *Alzheimers Res Ther* **14**, 26, doi:10.1186/s13195-021-00942-0 (2022).
- 3 Chang, C. H., Lin, C. H. & Lane, H. Y. Machine Learning and Novel Biomarkers for the Diagnosis of Alzheimer's Disease. *Int J Mol Sci* **22**, doi:10.3390/ijms22052761 (2021).
- 4 Wang, Y. Y. *et al.* The Proportion of APOE4 Carriers Among Non-Demented Individuals: A Pooled Analysis of 389,000 Community-Dwellers. *J Alzheimers Dis* **81**, 1331-1339, doi:10.3233/JAD-201606 (2021).

REVIEWERS' COMMENTS:

Reviewer #1 (Remarks to the Author):

I don't have further comment.

Reviewer #2 (Remarks to the Author):

I am happy with the revised version of the manuscript.

Reviewer #3 (Remarks to the Author):

They have addressed all my concerns

Point-by-Point Response to Reviewers' comments

Communications Medicine manuscript ID: COMMSMED-22-0360A.

Authors: Fengfeng Pan *et al.*

General response: We kindly thank editor and reviewers for time and effort you have dedicated to providing your valuable feedback on our manuscript. In response to editor and reviewers' comments, we have followed their suggestions to revise the manuscript carefully. We believe that this version of the manuscript is largely improved.

Point-by-point response to the comments of reviewer #1:

I don't have further comment.

Response: We really appreciate the time and effort you have dedicated to providing your valuable feedback on our manuscript.

Point-by-point response to the comments of reviewer #2:

I am happy with the revised version of the manuscript.

Response: We really appreciate the time and effort you have dedicated to providing your valuable feedback on our manuscript.

Point-by-point response to the comments of reviewer #3:

They have addressed all my concerns

Response: We really appreciate the time and effort you have dedicated to providing your valuable feedback on our manuscript.